# Variable Expressivity of the Beckwith-Wiedemann Syndrome in Four Pedigrees Segregating Loss-of-Function Variants of *CDKN1C*

**DOI:** 10.3390/genes12050706

**Published:** 2021-05-09

**Authors:** Angela Sparago, Flavia Cerrato, Laura Pignata, Francisco Cammarata-Scalisi, Livia Garavelli, Carmelo Piscopo, Alessandra Vancini, Andrea Riccio

**Affiliations:** 1Department of Environmental Biological and Pharmaceutical Sciences and Technologies (DiSTABiF), Università degli Studi della Campania “Luigi Vanvitelli”, 81100 Caserta, Italy; flavia.cerrato@unicampania.it (F.C.); laura.pignata@unicampania.it (L.P.); 2Pediatrics Service, Regional Hospital of Antofagasta, 5935 Antofagasta, Chile; francocammarata19@gmail.com; 3Medical Genetics Unit, Mother and Child Health Department, Azienda USL-IRCCS di Reggio Emilia, 42123 Reggio Emilia, Italy; livia.garavelli@ausl.re.it; 4Medical and Laboratory Genetics Unit, “Antonio Cardarelli” Hospital, 80131 Napoli, Italy; carmelo.piscopo@aocardarelli.it; 5Neonatal Intensive Care Unit, Maggiore Hospital, 40133 Bologna, Italy; alessandra.vancini@ausl.bologna.it; 6Institute of Genetics and Biophysics (IGB) “Adriano Buzzati-Traverso”, Consiglio Nazionale delle Ricerche (CNR), 80131 Napoli, Italy

**Keywords:** CDKN1C, Beckwith-Wiedemann syndrome, genomic imprinting, loss-of-function mutations, exomphalos

## Abstract

Beckwith-Wiedemann syndrome (BWS) is an imprinting disorder characterized by prenatal and/or postnatal overgrowth, organomegaly, abdominal wall defects and tumor predisposition. *CDKN1C* is a maternally expressed gene of the 11p15.5 chromosomal region and is regulated by the imprinting control region IC2. It negatively controls cellular proliferation, and its expression or activity are frequently reduced in BWS. In particular, loss of IC2 methylation is associated with *CDKN1C* silencing in the majority of sporadic BWS cases, and maternally inherited loss-of-function variants of *CDKN1C* are the most frequent molecular defects of familial BWS. We have identified, using Sanger sequencing, novel *CDKN1C* variants in three families with recurrent cases of BWS, and a previously reported variant in a woman with recurrent miscarriages with exomphalos. Clinical evaluation of the patients showed variable manifestation of the disease. The frameshift and nonsense variants were consistently associated with exomphalos, while the missense variant caused a less severe phenotype. Pregnancy loss and perinatal lethality were found in the families segregating nonsense mutations. Intrafamilial variability of the clinical BWS features was observed, even between siblings. Our data are indicative of severe BWS phenotypes that, with variable expressivity, may be associated with both frameshift and nonsense variants of *CDKN1C*.

## 1. Introduction

CDKN1C (also known as p57kip2) belongs to the CIP/KIP family of cyclin-dependent kinase (CDK) inhibitors and is required for embryonic development [1,2,3]. Although its major role is to negatively control cell proliferation, CDKN1C also regulates other cellular processes, including differentiation, migration, apoptosis, and senescence [4]. Two conserved domains, the CDK-binding domain (kinase inhibitory domain) and the proliferating cell nuclear antigen (PCNA)-binding domain, mediating the interaction with the replication fork, are located at the N-terminus and C-terminus of the protein, respectively (Figure 1A). Between these domains, the human CDKN1C contains a region of proline–alanine repeats (PAPA domain) of variable length, of which function remains unclear.

The gene encoding CDKN1C (MIM #600856) is regulated by genomic imprinting and is part of the imprinted gene cluster that in the human genome is located in the chromosomal 11p15.5 region. *CDKN1C* is expressed predominantly from the maternal allele and its imprinting is controlled by the *KCNQ1OT1*:TSS-DMR (also known as imprinting centre 2 or IC2), which is methylated on the maternal chromosome and overlaps the promoter of the long non-coding RNA *KCNQ1OT1*. On the paternal chromosome, IC2 is unmethylated and *CDKN1C* is repressed in cis by the *KCNQ1OT1* RNA.

In humans, *CDKN1C* deregulation contributes to the development of cancer and of some rare congenital diseases characterized by abnormal growth [4]. In particular, gain-of-function mutations of the PCNA binding domain increasing protein stability have been found in some rare cases of the growth restriction-associated IMAGE syndrome and Silver-Russell syndrome [5,6,7,8]. Conversely, impairment of the CDKN1C function or expression is the most common cause of the Beckwith-Wiedemann syndrome (BWS, OMIM #130650), an overgrowth disorder characterized by developmental abnormalities and predisposition to embryonal tumors.

All the BWS patients with a positive molecular diagnosis have mostly sporadic abnormalities in the 11p15.5 imprinted gene cluster [9]. Approximately 50% of these cases show loss of methylation of the IC2 that is associated with biallelic expression of *KCNQ1OT1* and suppression of the other imprinted genes of the IC2 domain, including *CDKN1C*. Loss-of-function genetic variants of *CDKN1C* are found in 5% of sporadic BWS cases and in 40% of the rarer familial cases. Another epigenetic defect of the BWS is gain of methylation of the *H19/IGF2*:IG-DMR (also known as Imprinting Centre 1 or IC1). This alteration occurs in 5–10% of BWS cases and results in biallelic activation of the fetal growth factor gene *IGF2* and suppression of noncoding RNA *H19* gene. Furthermore, mosaic paternal uniparental disomy of 11p15 (upd(11)pat) is found in 20% of the cases. Finally, rearrangements of the 11p15 region, mostly paternal duplications, have also been reported in <5% of BWS cases.

BWS is characterized by a broad phenotypic spectrum and a consensus scoring system has been developed to support its clinical diagnosis [9]. Two points have been assigned to each cardinal feature, such as macroglossia, exomphalos and lateralized overgrowth; and one point to each suggestive feature, such as neonatal overgrowth, ear lobe alterations and hypoglycemia (Appendix A). (Epi)genotype-phenotype correlation studies indicate that BWS patients with *CDKN1C* mutations have a higher rate of severe abdominal wall defects, notably exomphalos, but lower incidence of neonatal macrosomia and lateralized overgrowth [10]. A clinical scoring system has also been recently proposed for prenatally suspected BWS cases, which includes characteristic features detected by ultrasonography, such as exomphalos, mesenchymal dysplasia of the placenta and macroglossia (Appendix A) [11,12]. Prenatal diagnosis is indicated in the presence of a positive family history with a known molecular defect and can be confirmed by molecular tests [12].

Here, we describe loss-of-function variants of *CDKN1C*, which co-segregate with variable expressivity of the clinical features of BWS in four families. Prenatal or perinatal lethality was observed in the cases with the most deleterious mutations.

## 2. Materials and Methods

### 2.1. Patients

The subjects described in this study were referred to our laboratory with a clinical diagnosis of BWS or were women who had a prenatal clinical diagnosis suggestive of BWS in at least two conceptuses whose development ended in miscarriage. All the probands were born from unrelated parents.

### 2.2. Molecular Analyses

DNA was extracted from peripheral blood leukocytes using a conventional salting-out procedure. Coding exons and splice sites junctions of *CDKN1C* were amplified by PCR using the Accuprime GC-Rich Polymerase Kit (Life Technologies) in a SimpliAmp Thermal Cycler (Applied Biosystems). PCR conditions were optimized for the amplification of specific DNA fragments as described in Appendix A. PCR products obtained were then separated by gel electrophoresis and purified with the QIAquick Gel Extraction Kit (Qiagen). Sanger sequencing was performed by Eurofins Genomics and results analyzed using SnapGene Viewer. Variants were described in accordance with the Human Genome Variation Society recommendations, using the reference sequence NP_000067.1, encoding a 316 aa protein (isoform A).

## 3. Results

Eight unrelated BWS patients of our cohort were selected for *CDKN1C* sequencing analysis, because of the presence of the following molecular/clinical features: (1) normal methylation at the 11p15.5 IC1 and IC2; (2) wall defects (exomphalos, diastasis recti and umbilical hernia); (3) BWS features in other members of the family. We also included in this analysis two healthy women with a history of recurrent miscarriages with exomphalos. We identified four heterozygous variants of *CDKN1C* in distinct families. Consistent with the imprinted expression of this gene, the *CDKN1C* variants were maternally inherited by the affected subjects. Three variants were novel, not listed in gnomAD and dbSNP databases, and one was previously described in a familial case of BWS (Table 1).

Family 1 included two patients with typical BWS features. The older patient (F1_II-1) was a 16-year-old girl born with a small umbilical hernia that spontaneously closed, transient hypoglycemia, macroglossia and facial naevus simplex, that corresponds to a clinical score = 5 (Table 1). The 14-year-old younger brother (F1_II-3) was born with exomphalos, umbilical and inguinal hernias, macroglossia, facial naevus simplex and ear pits (Table 1). Overall, a clinical score of 7 was reached, also confirming the diagnosis of BWS in this case. In both patients, facial dysmorphisms and cardiovascular anomalies were also reported. In this family, we identified a missense variant (c.237G>C, p.Trp79Cys) in the N-terminal kinase inhibitor domain of CDKN1C (Figure 1A,B). Segregation analysis showed that the mother (F1_I-2) was an asymptomatic carrier who transmitted the variant to the affected children and the wild-type allele to her healthy son (F1_II-2) (Figure 1B). In silico analyses predict the variant as deleterious (SIFT, score 0.01), probably damaging (PolyPhen-2, score 1.00) and disease-causing (MutationTaster2, score 215). Moreover, Trp79 is conserved in primates and non-primates, as well as in marsupials and other vertebrates, suggesting a functional or structural role of this residue (Figure 1B).

In family 2, both the proband and his mother presented clinical manifestations of BWS (Figure 1C), although with distinct severity. At the time of the molecular diagnosis, the mother (F2_I-2) was a 38-year-old woman, who reported the presence of macroglossia and exomphalos at birth (score = 4, Table 1). Her son (F2_II-1) was a 19-year-old boy, who had typical BWS features at birth, including exomphalos, macroglossia, diastasis recti, facial naevus simplex, ear creases and hepatomegaly (score = 8, Table 1). In addition, he had facial dysmorphisms, ureterocele, meningocele and hypospadias. Clinical examination at 3 years revealed the presence of a cystic liver. This patient and his mother were carrier of an 11bp deletion in the *CDKN1C* coding region, c.627_637del, resulting in the frameshift variant p.Ala211Glyfs*26 in the central PAPA protein domain (Figure 1C). This variant leads to premature termination that is predicted to completely remove the C-terminal third of the protein (Figure 1A).

For the index patient of family 3 (F3_III-1), a prenatal diagnosis of exomphalos was postulated. Typical BWS features including transient hypoglycemia, macroglossia, facial naevus simplex and ear creases (score= 6) were observed at birth (Table 1). In addition, she had elevated birth length (>90th percentile), cleft palate, gut malrotation and facial dysmorphisms. Clinical examination at 5 years confirmed the presence of elevated height (119 cm, 97th percentile) and increased body weight (21.150 kg, 90–97th centile). Prenatal diagnosis of exomphalos was also obtained during the gestation of her younger sister (F3_III-2). At birth she also presented macroglossia, reaching a BWS score= 4 (Table 1). After three surgical interventions for the treatment of exomphalos, she sadly died at the age of 23 days. In this family, we identified a *CDKN1C* stop-gain variant (c.196C>T, p.Gln66*) that is predicted to truncate the kinase inhibitor domain and remove most of the protein (Figure 1A,D). As in the previous cases, the variant identified in the proposita (F3_III-1) was inherited from her mother (F3_II-1). Although a DNA sample of F3_III-2 was not available, it is very likely that she also inherited this *CDKN1C* mutation from her mother. The presence of the variant in the maternal grand-father (F3_I-1) confirms that the paternal transmission of loss-of-function *CDKN1C* mutations is associated with a healthy phenotype, because of genomic imprinting (Figure 1D).

Family 4 includes a 36-year-old woman (F4_I-2) with recurrent pregnancy loss (Figure 1E). No conception difficulty was reported for the couple, who had a healthy first child (F4_II-1). However, a second pregnancy was terminated after 12 weeks of gestation and the fetus presented exomphalos; a third pregnancy was spontaneously aborted for unknown reasons at 6 weeks of gestation; the last pregnancy also ended in a miscarriage at the 14th week of gestation. In this latter case, the fetus presented with exomphalos and adrenal cortex cytomegaly. A *CDKN1C* stop-gain variant (c.731C>A, p.Ser244*), predicted to remove the C-terminal PCNA binding domain, was identified in F4_I-2 (Figure 1A,E and Table 1). The DNA of the aborted fetuses was not available for testing.

## 4. Discussion

Both genetic and epigenetic mechanisms are responsible for BWS. Recurrence of BWS features in individuals of the same family, absence of epigenetic alterations of the 11p15.5 region and presence of severe abdominal wall defects is suggestive of familial BWS with *CDKN1C* mutations [10,13].

By selecting these features in our cohort, we were able to identify *CDKN1C* loss-of-function variants in four BWS families. The six patients and the two fetuses included in these families fulfilled the clinical criteria of BWS (Table 1, [9,12]). In agreement with previous reports, the most frequent cardinal feature of BWS observed in these patients was exomphalos, whereas lateralized overgrowth (also known as hemihypertrophy) that is usually associated with epigenetic mosaicism (epimutations or UPD) was absent [10]. It is possible that the cases with similar clinical features in which we could not identify *CDKN1C* variants may carry other genetic alterations affecting *CDKN1C* expression, such as promoter or enhancer variants that were not investigated.

Similar to other deleterious missense variants of *CDKN1C* described so far, p.Trp79Cys falls in the kinase inhibitor domain and involves a highly conserved amino acid with a possible functional or structural role [13,14]. Different severity levels of abdominal wall defects were observed in the two siblings of family 1, since the missense variant was associated with umbilical hernia in the proposita and exomphalos in her brother. This finding is in agreement with the milder phenotypic expression of missense mutations and with the variable expressivity of the BWS phenotype associated with *CDKN1C* variants, even among members of the same family [13].

Frameshift and nonsense mutations represent the majority of *CDKN1C* mutations found in BWS patients [13,14]. Frameshift variants are distributed along the entire protein length. Because of the repetitive sequence context at the end of the PAPA domain, p.Ala211Glyfs*26 probably originates by misalignment during DNA replication and subsequent deletion of 11 nucleotides. The proband of family 2 had the highest clinical score and several developmental abnormalities. In his mother, clinical manifestations, although severe, were less extended. We were not able to trace the variant’s segregation from the maternal grandparents and cannot exclude that p.Ala211Glyfs*26 arose de novo, likely on the maternal chromosome in F2_I-2. However, no evidence of mosaicism of the variant was evident from Sanger sequencing (Figure 1C). This suggests that both mother and son of family 2 presented a constitutional mutation of *CDKN1C* and that the milder BWS expressivity in the mother is a consequence of intrafamilial variability.

So far, nonsense variants have been found with lower frequency in the Kinase inhibitor domain and higher frequency between the PAPA domain and the PCNA binding domain [13,14]. Sense codons that are similar in sequence to stop codons, are more susceptible to nonsense mutations by single nucleotide substitutions. One of most frequent nucleotide substitutions giving rise to nonsense mutations in disease-causing genes is C to T transition in CAG codons [15]. Interestingly, most of the codons affected by stop-gain variants in the *CDKN1C* sequence are CAG (8/18, including the Gln66 mutated in family 3; Appendix A). Other less frequent nonsense variants of *CDKN1C* are G to T transversion in GAG codons (6/18) and C to A transversion in TCG codons (2/18, including the Ser244 mutated in family 4). It is worthwhile to note that the position of both CAG and GAG codons in the *CDKN1C* sequence might explain the bias in the distribution of nonsense variants (Appendix A).

Although localized in different positions of the CDKN1C protein, and, in principle, removing protein tracts of different length and alternative domains, the stop-gain variants of families 3 and 4 were responsible for similar dramatic clinical manifestations, including recurrence of exomphalos in multiple pregnancies and prenatal and perinatal lethality. Transcript harboring premature termination codons, caused by nonsense or frameshift mutations, can be degraded by nonsense-mediated mRNA decay (NMD), a cellular mechanism that limits the production of truncated proteins [16,17]. In particular, premature stop codons, more than 50–55 nucleotides upstream of the last exon–exon junction, typically trigger NMD [18]. Interestingly, p.Gln66* and p.Ser244* resided 624 and 89 nt upstream of the next exon 1–exon 2 junction, respectively. This suggests that, rather than inducing truncated proteins of different length, both mutations might induce NMD, causing complete absence of the CDKN1C protein.

In BWS cases with *CDKN1C* variants, miscarriages with exomphalos have been mostly reported in families segregating frameshift mutations [13,19]. Our study suggests that prenatal and perinatal lethality, possibly caused by severe exomphalos, can also be associated to nonsense *CDKN1C* variants. This is consistent with a previous study reporting on a woman carrying a p.Gln241* nonsense variant, with a history of three early pregnancy losses, a further miscarriage with exomphalos, and a son with severe BWS phenotype, including exomphalos [20].

p.Ser244* has already been reported in a familial case of BWS [13]. In that case (family 29), the variant was not associated with miscarriages and caused different abdominal wall defects in two patients: umbilical hernia in the mother and exomphalos in her son. The variable manifestation of BWS associated with constitutional *CDKN1C* variants might involve environmental factors or modifier genes influencing *CDKN1C* activity, although this issue remains to be characterized.

## 5. Conclusions

In summary, our findings confirm the high risk of recurrence and complete penetrance of BWS within pedigrees associated with maternal transmission of *CDKN1C* variants. Our data are indicative of a wide spectrum of expressivity that only in part depends on the type of mutation because both interfamilial and intrafamilial variability of clinical features were observed. We found that, similar to frameshift mutations, nonsense variants may be responsible for severe phenotypes and can be found in pedigrees showing recurrent miscarriages with exomphalos.

## Figures and Tables

**Figure 1 genes-12-00706-f001:**
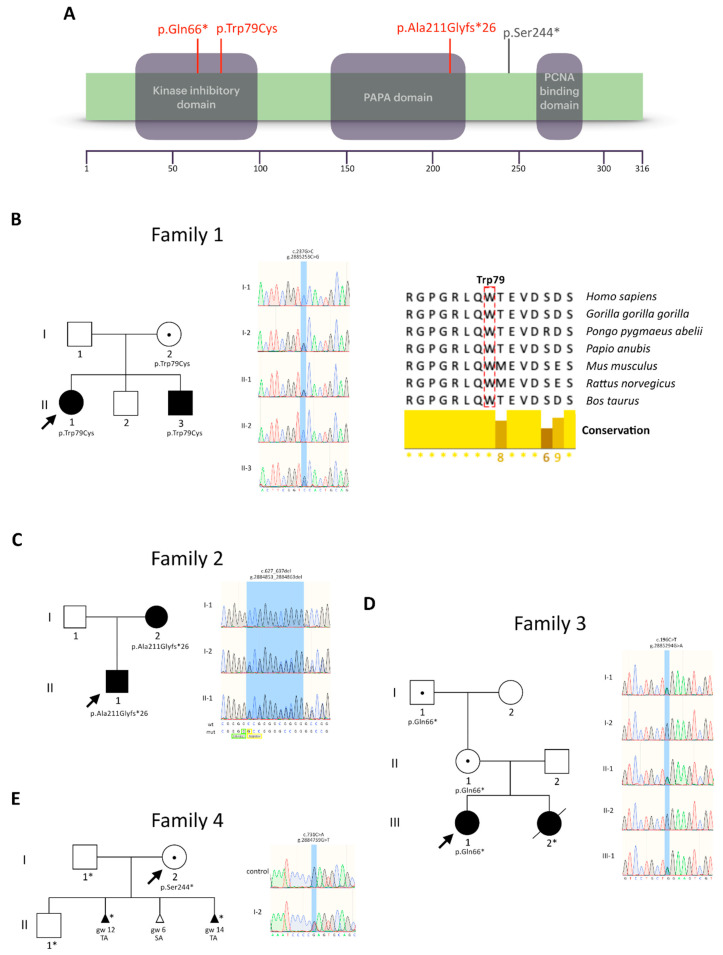
*CDKN1C* pathogenic variants in the four families under study. (**A**) Domain structure of the human CDKN1C (isoform A, NP_000067.1) depicting the position of the identified variants (novel variants in red; known variant in gray). (**B**) On the left, pedigree and DNA sequences showing the segregation of the variant in family 1. The patients or fetuses with BWS features are represented with black-filled symbols; the unaffected carriers of *CDKN1C* mutations with a dot in the middle of the symbol. The sequence variants are highlighted by a blue-shaded stripe in the electropherograms and their cDNA (NM_000076.2) and genomic (chr11, GRCh38/hg38) positions are indicated. On the right, evolutionary conservation of the tryptophan 79 (dashed in red) in primates and other mammals (Jalview software, V.2.11.1.3). (**C**–**E**) Pedigrees and DNA sequences showing the segregation of the variants in families 2–4. Asterisks indicate individuals unavailable for molecular analysis. Abbreviations: gw—weeks of gestation; TA—therapeutic abortion; SA—spontaneous abortion.

**Table 1 genes-12-00706-t001:** Beckwith-Wiedemann syndrome cases and *CDKN1C* pathogenic variants identified in this study.

Individual	Anthropometric Data at Birth (gw, W, L)	Clinical Features									Clinical Score	Protein Change (NP000067.1, 316aa)	cDNA Change (NM_000076.2); Genomic Location (Chr 11, GRCh38/hg38)	dbSNPs	gnomAD	Ref.
		Macroglossia	Exomphalos	Facial Naevus Simplex	Ear Signs	Transient Hypoglycaemia	Hepatomegaly	Umbilical Hernia	Diastasis Recti	Adrenal Cortex Cytomegaly						
F1_II-1	40, 3160 g, 50 cm	+	−	+	−	+	−	+	−	−	5 ^a^	p.Trp79Cys	c.237G>C g.2885253C>G	nr	nr	-
F1_II-3	38, 3000 g, 48.5 cm	+	+	+	+	−	−	+	−	−	7 ^a^					
F2_I-2	-	+	+	−	−	−	−	−	−	−	4 ^a^	p.Ala211Glyfs*26	c.627_637del g.2884853_2884863del	nr	nr	-
F2_II-1	37, 3270 g, 49 cm	+	+	+	+	−	+	−	+	−	8 ^a^					
F3_III-1	38, 3400 g, 53 cm *	+	+	+	+	+	−	−	−	−	6 ^a^	p.Gln66*	c.196C>T g.2885294G>A	nr	nr	-
F3_III-2 ^†^	38, 3500 g, 50 cm	+	+	−	−	−	−	−	−	−	4 ^a^					
F4_fetus 1	-	−	+	−	−	−	−	−	−	−	6 ^b^	p.Ser244*	c.731C>A g.2884759G>T	rs483352993	nr	[13]
F4_fetus 3	-	−	+	−	−	−	−	−	−	+	6 ^b^					

a: clinical scoring system of BWS patients (cut-off ≥ 4 for clinical diagnosis [9]); b: prenatal scoring system of suspected BWS cases (cut-off ≥ 3 for molecular testing [12]); nr: not reported; gw: gestational weeks; W: weight; L: length; *: 90–97th centile; †: deceased perinatally.

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
