# Peer review of "Variable Expressivity of the Beckwith-Wiedemann Syndrome in Four Pedigrees Segregating Loss-of-Function Variants of CDKN1C"

_genes, 2021, doi:10.3390/genes12050706_

Round 1

Reviewer 1 Report

The paper describes the identification of three novel protein-truncating mutations in CDKN1C in eight unrelated families with recurrent Beckwith-Wiedemann Syndrome (BWS). The findings are novel and the study is important. My comments below are minor and meant to help the authors clarify the paper and make it easier for the readers.

-line 42, “A cyclin-42 CDK binding domain, involved i….” This sentence is very long and unclear. It needs to be reformulated.

-line 117, “familiarity for BWS features”. The term familiarity is not correct here. It is better to say familial or the presence of some BWS features in other family members.

-lines 122 & 253 and, “p.S244X has been already reported in a BWS pedigree…”  Better to say in a familial case of BWS. A pedigree does not mean that there are two affected members.

-line 154, “ ….carrier of a 11bp deletion of the CDKN1C gene, responsible for the frameshift variant p.A211GfsX26 in the central….”. The “p.A211GfsX26” is the protein name of the observed 11bp deletion. Better to say, an 11-bp deletion in the CDKN1C coding region followed by comma, the cDNA name of the mutation then comma and the protein name of the mutation. Also, because all the identified mutations are in the coding regions, it is better to provide the cDNA names of the mutations instead of the genomic DNA names in figure 1 and in the text.

-line 155, “This variant leads to a premature termination that might completely removes the C-terminal”. It is not that it might remove, it is predicted to remove …..This is the only consequence of such mutation  and not one of many possibilities.

-line 169, “ we identified a CDKN1C stop-gain variant  (p.Q66X), that could truncate the Kinase Inhibitor Domain”. Again, it is not that could, this is a protein truncation (again not an option), one can say “…..(p.Q66X), that truncates the protein in the middle of the Kinase Inhibitor domain.

Author Response

Response to Reviewer 1 comments

The paper describes the identification of three novel protein-truncating mutations in CDKN1C in eight unrelated families with recurrent Beckwith-Wiedemann Syndrome (BWS). The findings are novel and the study is important. My comments below are minor and meant to help the authors clarify the paper and make it easier for the readers.

Point 1: line 42, “A cyclin-42 CDK binding domain, involved i….” This sentence is very long and unclear. It needs to be reformulated.

Response 1: As suggested by the reviewer, we have modified the sentences, lines 42-50 (revised ms): “”Two conserved domains, the CDK-binding domain (Kinase inhibitory domain) and the Proliferating Cell Nuclear Antigen (PCNA)-binding domain, mediating the interaction with the replication fork, are located at the N-terminus and C-terminus of the protein, respectively (Figure 1A). Between these domains, the human CDKN1C contains a region of proline-alanine repeats (PAPA domain) of variable length, whose function remains unclear.

Point 2: line 117, “familiarity for BWS features”. The term familiarity is not correct here. It is better to say familial or the presence of some BWS features in other family members.

Response 2: The sentence “familiarity for BWS features” had been replaced with “BWS features in other members of the family”, line 221 (revised ms).

Point 3: lines 122 & 253 and, “p.S244X has been already reported in a BWS pedigree…”  Better to say in a familial case of BWS. A pedigree does not mean that there are two affected members.

Response 3: Done, lines 226 and 533 (revised ms).

Point 4: line 154, “ ….carrier of a 11bp deletion of the CDKN1C gene, responsible for the frameshift variant p.A211GfsX26 in the central….”. The “p.A211GfsX26” is the protein name of the observed 11bp deletion. Better to say, an 11-bp deletion in the CDKN1C coding region followed by comma, the cDNA name of the mutation then comma and the protein name of the mutation. Also, because all the identified mutations are in the coding regions, it is better to provide the cDNA names of the mutations instead of the genomic DNA names in figure 1 and in the text.

Response 4: To address this important point of reviewer, we have modified the sentence as suggested, line 322 (revised ms). Moreover, for all mutations we have added the cDNA name in the main text (lines 306, 335, 349 of revised ms), in Table 1 and in Figure 1.

Point 5: line 155, “This variant leads to a premature termination that might completely removes the C-terminal”. It is not that it might remove, it is predicted to remove …..This is the only consequence of such mutation  and not one of many possibilities.

and

Point 6: line 169, “ we identified a CDKN1C stop-gain variant  (p.Q66X), that could truncate the Kinase Inhibitor Domain”. Again, it is not that could, this is a protein truncation (again not an option), one can say “…..(p.Q66X), that truncates the protein in the middle of the Kinase Inhibitor domain.

Response 5-6: In agreement with the concern of the reviewer, “might” and “could” have been replaced with “is predicted to …”, lines 324, 336 and 349 (revised ms).

Reviewer 2 Report

Thank you for the opportunity to review the manuscript “Variable expressivity of the Beckwith-Wiedemann Syndrome in four pedigrees segregating loss-of-function variants of CDKN1C ” by Angela Sparago et al. Beckwith-Wiedemann Syndrome (BWS) is a rare, genetically and clinically heterogeneous disorder. The authors report novel variants of CDKN1C in three families and one previously reported variant in a woman with recurrent miscarriages with exomphalos. Therefore, the manuscript reveals a novel genetic background of BWS with a detailed clinical synopsis of the patients. I find the report interesting and valuable for both geneticists and clinicians. 

However, there are some points that need to be improved.

  1. Abstract, line 21: “pre-postnatal overgrowth” is unclear, I would suggest writing “prenatal and/or postnatal overgrowth”.
  2. Introduction: although the manuscript regards CDKN1C I would broaden the background of BWS and add a paragraph about other possible genes involved in the syndrome. I think it is necessary to understand the variety of types of genetic diagnoses.
  3. Lines 70-74: in my opinion it would be more clear to put the clinical features and scoring in a Table. It would make it easier to follow different clinical symptoms in presented patients.
  4. Results: please make them more concise and reconsider putting the birth anthropometric data in a Table. Do we have more details regarding neonatal period, such as Apgar score, need of intensive neonatal care?
  5. CDKN1C should be written in italics (Table 1, Results).

Author Response

Response to Reviewer 2 comments

Thank you for the opportunity to review the manuscript “Variable expressivity of the Beckwith-Wiedemann Syndrome in four pedigrees segregating loss-of-function variants of CDKN1C ” by Angela Sparago et al. Beckwith-Wiedemann Syndrome (BWS) is a rare, genetically and clinically heterogeneous disorder. The authors report novel variants of CDKN1C in three families and one previously reported variant in a woman with recurrent miscarriages with exomphalos. Therefore, the manuscript reveals a novel genetic background of BWS with a detailed clinical synopsis of the patients. I find the report interesting and valuable for both geneticists and clinicians. 

However, there are some points that need to be improved.

Point 1: Abstract, line 21: “pre-postnatal overgrowth” is unclear, I would suggest writing “prenatal and/or postnatal overgrowth”.

Response 1: Done, line 21 (revised ms).

Point 2: Introduction: although the manuscript regards CDKN1C I would broaden the background of BWS and add a paragraph about other possible genes involved in the syndrome. I think it is necessary to understand the variety of types of genetic diagnoses.

Response 2: As required, the background of BWS has been completely rewritten in this paragraph of Introduction, lines 66-77 (revised ms):

“All the BWS patients with a positive molecular diagnosis have abnormalities in the 11p15.5 imprinted gene cluster and are mostly sporadic [9]. Approximately 50% of these cases show loss of methylation of the IC2 that is associated with biallelic expression of KCNQ1OT1 and suppression of the other imprinted genes of the IC2 domain, including CDKN1C. Loss-of-function genetic variants of CDKN1C are found in 5% of sporadic BWS cases and in 40% of the rarer familial cases. Another epigenetic defect of the BWS is gain of methylation of the H19/IGF2:IG-DMR (also known as Imprinting Centre 1 or IC1). This alteration occurs in 5-10% of BWS cases and results in biallelic activation of the fetal growth factor gene IGF2 and suppression of noncoding RNA H19 gene. Furthermore, mosaic paternal uniparental disomy of 11p15 (upd(11)pat) is found in 20% of the cases. Finally, rearrangements of the 11p15 region, mostly paternal duplications, have been also re-ported in <5% of BWS cases.”

Point 3: Lines 70-74: in my opinion it would be more clear to put the clinical features and scoring in a Table. It would make it easier to follow different clinical symptoms in presented patients.

Response 3: A table showing cardinal, suggestive and supportive features of BWS and their relative scoring has been added as Supplementary Table 1 (revised ms).

Point 4: Results: please make them more concise and reconsider putting the birth anthropometric data in a Table. Do we have more details regarding neonatal period, such as Apgar score, need of intensive neonatal care?

Response 4: As required, the Results have been shortened and the anthropometric data of the patients at birth added to Table 1. All available clinical data regarding the neonatal period have been described.

Point 5: CDKN1C should be written in italics (Table 1, Results).

Response 5: Done.

English

We have checked the manuscript for correct use of English language.
